# Vibrational Spectroscopy Coupled to a Multivariate Analysis Tiered Approach for Argentinean Honey Provenance Confirmation

**DOI:** 10.3390/foods9101450

**Published:** 2020-10-13

**Authors:** Tito Damiani, Rosa M. Alonso-Salces, Inés Aubone, Vincent Baeten, Quentin Arnould, Chiara Dall’Asta, Sandra R. Fuselli, Juan Antonio Fernández Pierna

**Affiliations:** 1Department of Food and Drugs, University of Parma, Viale delle Scienze 17/A, 43124 Parma, Italy; tito.damiani@studenti.unipr.it; 2Grupo de Investigación Microbiología Aplicada, Centro de Investigación en Abejas Sociales, Facultad de Ciencias Exactas y Naturales, Universidad Nacional de Mar del Plata, Dean Funes B7602AYL, Mar del Plata 3350, Argentina; rosamaria.alonsosalces@gmail.com (R.M.A.-S.); inesaubone@gmail.com (I.A.); sandra.fuselli@gmail.com (S.R.F.); 3Departamento de Biología, CONICET, Facultad de Ciencias Exactas y Naturales, Universidad Nacional de Mar del Plata, Funes 3350, Mar del Plata 7600, Argentina; 4Quality and Authentication of Products Unit, Knowledge and Valorization of Agricultural Products Department, Walloon Agricultural Research Centre (CRA-W), Chée de Namur, 24, 5030 Gembloux, Belgium; v.baeten@cra.wallonie.be (V.B.); q.arnould@cra.wallonie.be (Q.A.); j.fernandez@cra.wallonie.be (J.A.F.P.); 5Comisión de Investigaciones Científicas (CIC), La Plata, Argentina Camino General Belgrano 526, La Plata 1900, Argentina

**Keywords:** honey, vibrational spectroscopy, geographical origin, chemometrics, data fusion

## Abstract

In the present work, the provenance discrimination of Argentinian honeys was used as case study to compare the capabilities of three spectroscopic techniques as fast screening platforms for honey authentication purposes. Multifloral honeys were collected among three main honey-producing regions of Argentina over four harvesting seasons. Each sample was fingerprinted by FT-MIR, NIR and FT-Raman spectroscopy. The spectroscopic platforms were compared on the basis of the classification performance achieved under a supervised chemometric approach. Furthermore, low- mid- and high-level data fusion were attempted in order to enhance the classification results. Finally, the best-performing solution underwent to SIMCA modelling with the purpose of reproducing a food authentication scenario. All the developed classification models underwent to a “year-by-year” validation strategy, enabling a sound assessment of their long-term robustness and excluding any issue of model overfitting. Excellent classification scores were achieved by all the technologies and nearly perfect classification was provided by FT-MIR. All the data fusion strategies provided satisfying outcomes, with the mid- and high-level approaches outperforming the low-level data fusion. However, no significant advantage over the FT-MIR alone was obtained. SIMCA modelling of FT-MIR data produced highly sensitive and specific models and an overall prediction ability improvement was achieved when more harvesting seasons were used for the model calibration (86.7% sensitivity and 91.1% specificity). The results obtained in the present work suggested the major potential of FT-MIR for fingerprinting-based honey authentication and demonstrated that accuracy levels that may be commercially useful can be reached. On the other hand, the combination of multiple vibrational spectroscopic fingerprints represents a choice that should be carefully evaluated from a cost/benefit standpoint within the industrial context.

## 1. Introduction

According to the European Union Council Directive 2001/110/EC [1] and FAO/WHO Codex Alimentarius [2], honey is defined as the natural substance produced by *Apis mellifera* bees from plant nectar or excretions of plant-sucking insects. As a relative expensive food commodity, honey is known to be highly vulnerable to adulteration with the main concern historically being its dilution with cheaper sugars and/or syrups. Nowadays, the premium price usually commanded by mono-floral and mono-geographic products encourages other fraud practices such as false origin labelling or misdescription [3].

Reliable analytical methods for the honey authenticity assessment are highly claimed and lot of research has been undertaken in this field. Botanical origin is traditionally confirmed by melissopalynology, a microscopy-based qualitative and quantitative characterization of pollen [4]. This technique has been tested also for geographical discrimination purposes, but its application suffers from methodological shortages and limitations [3,5]. Consequently, novel alternative approaches have been proposed, including those based on mass spectrometry, vibrational spectroscopy and molecular biology [6,7,8]. The targeted quantification of specific compounds indicative for certain properties and/or origin would represent the most straightforward approach for food authentication; the comparison of the measured parameter with a control limit would empower the direct assessment of the product compliance and might also be used for forensic purposes [9]. However, finding reliable authenticity markers for honey’s botanical/geographical origin proved to be a hard task due to the number of factors affecting its chemical composition (e.g., beekeeping technique, harvest and storage environmental conditions, etc.). In addition, the analytical output may strongly depend on the adopted sample preparation procedure, hindering the data comparison and interpretation [10].

Over the past few years, new food testing strategies based on the so-called fingerprinting approaches have been introduced. The intrinsic aim of food fingerprinting is the non-targeted detection of as many features as technically possible, by means of high-throughput techniques, to gain a comprehensive insight into the sample composition. The recorded output consists of multidimensional datasets which, beside relevant information, may also contain unintended systematic and random variation. For this reason, mathematical and statistical tools (multivariate analysis/chemometrics) constitute an integral part of the fingerprinting workflow for the extraction of meaningful information from the raw data [9]. A review of the main fingerprinting technologies has been published by Ellis et al., with particular interest toward vibrational spectroscopy techniques, namely Raman, near- and mid-infrared spectroscopy [11]. These platforms offer non-destructive and cost-effective solutions to get quick spectral information about the tested material; the easy-of-use and potential on/in-line implementation represent further advantages over traditional methods that contributed to their spread in virtually all branches of agricultural and food industries [12].

In the honey authenticity field, the potential of vibrational spectroscopy coupled to multivariate data analysis to confirm the product’s claimed provenance [6,13,14,15,16,17] and/or botanical origin [18,19,20] has been widely investigated. Most of the published works are represented by truthful feasibility studies that demonstrated the capability of the employed technologies to capture differences between the analysed honey samples. To this end, discriminant analysis (DA) techniques have been used to develop supervised classification models that would correctly assign each sample to its belonging class. However, in real-world authentication contexts, no information is normally available about the alternative classes to which the tested item may belong. Indeed, the goal is typically to establish whether the analysed sample is compliant or not with a defined reference standard. For these reasons, DA methods have been defined inappropriate for solving food authenticity problems by several authors [21,22,23]. In contrast, one-class classifier (OCC) approaches should be preferred. Furthermore, the sample collection in the above-mentioned studies was most often limited to 1–2 years, thus hardly representative of the potential seasonal variability. This has certainly posed some limitations for a solid validation of the achieved classification results. As a matter of fact, the adaption of existing models to new harvests is a problem scarcely addressed in pilot studies, usually due to the limited samples and/or resources availability. Nevertheless, it represents an essential challenge to be faced for a relevant implementation of non-targeted fingerprinting approaches in routine analysis [24].

The present work deals with the geographical origin discrimination of Argentinian honeys. Multifloral honeys were collected from three main honey-producing Argentinian provinces (i.e., Buenos Aires, Catamarca, Misiones) and the sampling was repeated over four harvesting seasons, from 2014 to 2017. Each sample was fingerprinted by near-infrared (NIR), Fourier-transform mid-infrared (FT-MIR) and Raman (FT-Raman) spectroscopy. The main intention was not the development of a multivariate model able to correctly classify the analysed samples according to their provenance. Rather, the aim was to use this survey as a case study to compare the capabilities of the employed spectroscopic techniques as fast screening platforms for honey authentication purposes. In order to further improve the results obtained by the individual techniques, different data fusion strategies were attempted. Finally, the best-performing solution (i.e., either individual or fused data) was further modelled using an OCC approach with the purpose of reproducing a food authentication scenario and establish whether commercially useful accuracy levels can be reached. All the developed classification models underwent to a “year-by-year” validation strategy that enabled a sound assessment of their long-term robustness and excluded any issue of model overfitting.

## 2. Materials and Methods

### 2.1. Sample Collection

Authentic and traceable multifloral honey samples were collected from three main honey-producing provinces of Argentina: Buenos Aires (BA), Catamarca (Cat) and Misiones (Mis) (Appendix A), within the framework of the Argentinean National Projects PICT 3264/2014 and PICT 0774/2017, following the instructions depicted on the Projects’ analytical plan, and used for the scope of the present study. The samples (about 1 Kg of raw honey each) were provided directly by beekeepers and/or honey producer cooperatives along with farming information: harvest date and conditions, declared botanical origin, field or hive address and GPS coordinates, agricultural system, treatments, etc. The honeys were harvested between April and August and the sampling was repeated over four harvesting seasons (i.e., 2014, 2015, 2016 and 2017). Collected information on honey samples are to be considered part of the above-mentioned projects, and may be available upon request according to the data protection policy.

From here on, the sample batches (i.e., honeys from each harvest) are referred to as HN2014, HN2015, HN2016, HN2017, respectively. The total number of samples was *n* = 502 and an overview of the sample set is given in Appendix A. After collection, the honeys were stored in screw-capped glass containers, in the dark and at 4 °C, until analysis.

### 2.2. Instrumental Analysis

All the collected samples were fingerprinted by means of FT-MIR, FT-Raman and NIR. After the collection, each sample batch (i.e., harvest) was scanned over a 14-day period. Prior to the analysis, the honeys were incubated at 40 °C and manually stirred in order to dissolve any crystalline residue material. Quality control materials were scanned throughout the whole analysis in order to monitor potential batch-to-batch instrumental drift.

FT-MIR spectra were recorded in attenuated total reflection (ATR) mode, on a Vertex 70 FT-IR spectrometer (Bruker, Billerica, MA, USA), equipped with a Globar source, a DLaTGS detector and a Golden Gate ATR cell (Specac Ltd., Orpington, UK). Analyses were carried out in triplicate, placing the honey samples directly on the ATR crystal. All the spectra were computed at 4 cm^−1^ resolution, across the spectral range 4000–600 cm^−1^ and averaging a total of 64 scans. Data export was performed by Opus 7.2 software (Bruker).

FT-Raman spectra were collected on a Vertex 70 equipped with the RAM II add-on module (Bruker), a laser source emitting at 1064 nm and a Ge^(418-T/R)^ detector cooled by liquid N_2_. The laser power was set to 0.8 W. Honey samples were placed in a glass tube and analyzed in duplicate, across the spectral range 3600–0 cm^−1^, at a nominal resolution of 4 cm^−1^. Each spectrum was obtained by averaging 128 scans and exported with Opus 7.2 software (Bruker).

NIR spectroscopic analysis was performed on an XDS Vis/NIR spectrometer (FOSS Analytical, Hilleroed, Denmark) equipped with a tungsten halogen lamp and a dual detector Si (400–1100 nm) and PbS (1100–2500 nm). The spectra were recorded in transflectance mode, directly depositing the honey on the golden reflector. The analysis ran in duplicate and a total of 16 scans were averaged for each spectrum, at a nominal resolution of 2 nm, across the spectral range 400–2500 nm. Signal acquisition and export were performed by ISIscan software (FOSS Analytical).

### 2.3. Statistical Data Analysis

All the chemometric computations were carried out using Matlab v2019b (The Mathworks, Inc., Natick, MA, USA) and the PLS Toolbox (Eigenvector Research, Inc., Manson, WA, USA).

#### 2.3.1. Data Preprocessing

Prior to any exploratory or classification analysis, spectral preprocessing was applied to reduce the impact of unwanted sources of variability on the overall signal, thus highlighting the chemical information contained in the spectra. Different algorithms for spectral pretreatment, namely 1st and 2nd order derivative according to the Savitzky–Golay method (S-G), multiplicative scatter correction (MSC) and standard normal variate (SNV), were tested both on their own and in combination. The SNV and MSC are both designed to remove from reflectance spectra part of the variability that may be caused by scattering effects. In many cases, these two spectral pretreatment produced very similar results, so that they are widely regarded as exchangeable [25]. S-G derivative filter emphasizes band width, position, and separation while simultaneously reducing baseline and background effects [26].

#### 2.3.2. Unsupervised Pattern Recognition

After the preprocessing, principal component analysis (PCA) was performed as exploratory data analysis for the detection of evident outlying samples and/or potential data structures in a reduced-dimension space. The underlying concept of the PCA is to decrease the dimensionality of a dataset containing a large number of interrelated variables, while retaining as much as possible of the initial data variation. The original descriptors are “compressed”, through linear combination, into a new set of uncorrelated variables (i.e., principal components, PCs), which point in the directions of maximal variance. The so-called scores and loadings constitute the main output of the PCA. The scores represent the newly computed latent variables onto which the objects are projected, therefore they can be interpreted in exactly the same way as any other variable. On the other hand, the loadings are the weights given to the original variables during the computation of the PCs; thus, they determine what a PC represent. Both scores and loadings can be graphically plotted as line or scatter plots [27].

#### 2.3.3. Supervised Pattern Recognition and Validation Strategy

The employed spectroscopic techniques were compared on the basis of the classification performance achieved under a supervised chemometric approach, by using partial least squares discriminant analysis (PLS-DA) as classification algorithm. PLS-DA is arguably the most widely used DA technique, particularly suitable for dealing with data matrices characterized by a large number of highly correlated variables, such as spectroscopic data. PLS-DA can be regarded as a linear two-class classifier, although extension to more than two groups is also possible. The method aims to find a linear decision function(s) that divides the multidimensional variable space into as many regions as the number of classes. The objects are then projected onto lines orthogonal to this function and their distance along this discriminator is considered as discriminant score [28].

Binary PLS-DA models were generated on each data block, considering two geographical regions at once (i.e., BA-Mis, BA-Cat, Cat-Mis). At first, the models were built including all the harvesting seasons and optimized through “leave-one-out” cross-validation. Afterwards, the so-called receiver operating characteristic (ROC) curves were derived. ROC curves are widely used in many application fields as they allow a straightforward comparison of binary classifier systems. In the multivariate case, the curves are built varying the criterion threshold at which the classification is performed. Model’s sensitivity (i.e., fraction of compliant objects correctly accepted) and specificity (i.e., fraction of alien objects correctly rejected) are computed at each step and graphically represented in a two-axis Cartesian plot, in which 1-specificity is usually reported on the x-axis against the sensitivity on the y-axis. Experimental outcomes are connected by a line that constitutes the ROC curve. The area under the curve (AUC) is often used as summary measure of the general discrimination quality of the model. Intuitively, the larger the AUC, the higher the model classification ability. The ideal situation would be with both sensitivity and specificity equal to 1, which corresponds to a curve passing through the top-left corner of the graph and an AUC=1; in contrast, a curve lying on the diagonal bisector (corresponding to an AUC=0.5) suggests no discrimination [23].

Since ROC curves were built upon a cross-validation procedure, which may be prone to overfitting, the results reliability was ensured by the following validation strategy. At first, models were trained on the HN2014 and the provenance of HN2015 was predicted. Afterwards, the training set was augmented with the HN2015 samples and the models, upon re-optimization, were applied for the prediction of HN2016 provenance. As final step, HN2014, HN2015 and HN2016 were included in the training set and the HN2017 samples were classified. In this manner, the whole process involved three external validation steps independent of each other; thus, it can be considered much more reliable than a cross-validation approaches [29]. The validation scheme is summarized in Appendix A.

#### 2.3.4. Data Fusion

Since each honey was fingerprinted by three spectroscopic techniques, three different data matrices for the same sample set were obtained. The process of integrating multiple data blocks into a single global model is called data fusion (DF) and can lead to improvements of the classification accuracy respect to the individual data sources. Essentially, three DF strategies have been proposed in literature according to the degree of information merged: low, mid- and high-level data fusion (LL-, ML- and HL-DF, respectively). In LL-DF, data from all sources are simply concatenated column-wise into a single array. The merged matrix is then processed by the desired chemometric technique. ML-DF operates in a similar way, but relevant features are previously extracted from each data sources, separately. These features can be original descriptors identified as relevant or, more commonly, latent variables (e.g., PCA scores). The so-extracted variables are then concatenated prior to the multivariate data analysis. Lastly, in the HL-DF, separate models are built on the individual data blocks and the fusion occurs at the decision level, i.e., the individual predictions are integrated into a single final response. A more detailed description of DF methodologies employed in food and beverage authentication can be found in [30].

In the present study, LL-, ML- and HL-DF were attempted for the HN2017 prediction (i.e., last step of the year-by-year validation) with the aim of improving the performance of the single techniques. Briefly:

LL-DF: FT-MIR, FT-Raman and NIR data blocks consisted of 1349, 3009 and 751 variables, respectively. Each dataset was preprocessed according to its optimal spectral pretreatment prior to the concatenation. As a result, each sample was described by 5109 predictors. Autoscaling was applied to the fused matrix before further modelling;

ML-DF: PCA was separately performed on the training set of each data block. HN2017 objects were projected onto the PCs space so that both training and test sets were described by the same (latent) variables. Thereafter, PCA scores obtained from the individual blocks were merged and used for subsequent modelling;

HL-DF: The provenance of HN2017 samples was separately predicted carrying out PLS-DA on the individual data blocks as described in Section 2.3.3. Therefore, three column vectors containing the predicted classes were obtained and merged into a single array. The final decision on the class membership was made upon majority vote criterion.

#### 2.3.5. Soft Independent Modelling of Class Analogy

Soft independent modelling of class analogy (SIMCA) was the first class-modelling method introduced in the literature. It is a non-probabilistic distance-based modelling which relies on the assumption that the main systematic variability of the class of interest can be captured by a PCA model of appropriate dimensionality. The results of the PCA decomposition of the target category are used to define the so-called SIMCA inner space. At this point, the membership of the tested objects is decided on the basis of some statistical criterion for outlier detection. A comprehensive tutorial of SIMCA, and OCC methods in general, is provided in [23].

In the present study, being the most represented within the sample set, BA was set as target class whereas Catamarca and Misiones honeys were used as alien objects to challenge the model. The “degree of outlyingness” with respect to the target category was computed as combination of the Mahalanobis distance to the center of the inner space (*T*^2^) and the orthogonal distance (*Q*). For multivariate models whose assignation rule is based on the combined *T*^2^-*Q* distances, the classification outcome can be graphically represented in a Cartesian plot reporting the *T*^2^ and *Q* of the tested objects on the *x*- and *y*-axis, respectively. Roughly, the further from the origin (down-left corner) the sample is, the higher is its degree of outlyingness.

The same validation strategy described in Section 2.3.3 was adopted to ensure the reliability of the obtained classification results.

## 3. Results

### 3.1. Data Exploration

Prior to any chemometric manipulation, the recorded raw spectra of all honey samples were plotted and visually inspected (Appendix A). While very consistent FT-MIR and NIR spectra were obtained, FT-Raman spectra exhibited evident baseline drift, likely due to fluorescence phenomena. Therefore, the optimal combination of spectral filters and/or mathematical preprocessing was found to be SNV + S-G derivative (1st order derivative, 2nd order polynomial, 9 points window) + Mean centering for FT-MIR and NIR spectra, whereas a baseline correction step (manually-selected points, 3rd order polynomial, 5 regions) prior to SNV + Mean centering was included in the FT-Raman data preprocessing workflow.

As explained in Section 2.1, each sample batch (i.e., harvest) was scanned within 14 days after the collection. However, the analysis of the whole sample set was performed over a 4-years period. Therefore, the spectra recorded from the quality control materials were both visually examined and inspected through PCA in order to reveal any batch-to-batch instrumental drifts. No substantial spectral differences and/or separation in the scores plot were observed further to the application of SNV as data pretreatment (data not shown).

Once the data consistency had been ensured, PCA was carried out on the preprocessed honey spectra. The first three PCs accounted for more than 87% of the total variance in all the datasets. Regardless of the used platform, the PC1 vs. PC2 scores plot highlighted a noteworthy separation between BA and Mis honeys, whereas Cat samples were more scattered (Figure 1). Visual examination of higher order PCs did not reveal any greater degree of separation. Here too, no apparent clustering related to the harvesting year was noticed.

As denoted by the PC1 and PC2 loadings (Appendix A), the variables that shown the highest relevance in the PCs definition all corresponded to chemically meaningful spectral intervals. Specifically, most of the dispersion among the samples is explained by the wavelength range 1500–600 cm^−1^ for FT-MIR, 3000–2900 and 1500–0 cm^−1^ for FT-Raman, 480–600 and 1850–2500 nm for NIR.

According to previous reports, carbohydrate moieties are chiefly responsible for absorptions in these ranges of the honey spectra [14,15,31]. Noisy and/or uninformative spectral regions, i.e., CO_2_ band and flat regions, were excluded from the subsequent data treatment. As a result, the considered wavelength ranges were, respectively, 3800–2400 cm^−1^ and 1990–600 cm^−1^ for FT-MIR, 3600–2500 cm^−1^ and 1800–0 cm^−1^ for FT-Raman; 400–700 nm and 1300–2500 nm for NIR (Appendix A).

Band assignment was not the main goal of the study as the general tendency in fingerprinting methods is to use the entire spectra in the multivariate data analysis [32]. Nevertheless, description of the main peaks/bands responsible for the sample discrimination might be helpful for future research. Therefore, illustration of the statistically-significant spectral signals and of the three datasets has been reported in Appendix A (Appendix A). Furthermore, assignment of the relevant peaks/bands was carried out based on the literature [6,13,20,31,33,34,35,36,37].

### 3.2. Techniques Comparison under a Supervised Chemometric Approach

Classification outcomes provided by the individual spectroscopic techniques, as well as the fused datasets, are summarized in this section. ROC curves were constructed as described in Section 2.3.3 and graphically reported in Figure 2.

As expected from the unsupervised pattern recognition, better results were reached in the discrimination of Mis honeys (i.e., BA-Mis and Cat-Mis models). In particular, the BA-Mis model produced nearly perfect classification, with AUC always above 0.99 regardless the spectroscopic technique. In contrast, the BA-Cat model provided slightly lower AUC, ranging from 0.88 (NIR) to 0.93 (FT-MIR), perhaps due to unbalanced number of samples available. Concerning the inter-platforms comparison, FT-MIR provided yielded the largest AUC in all the binary models, while the lowest score was always obtained by FT-Raman spectroscopy.

The results of the validation procedure are summarized as correct classification rates (i.e., ratio between correctly classified and total tested objects, CCRs) in Table 1. For purposes of presentation, only the scores provided by FT-MIR data were reported, while FT-Raman and NIR data are available in Appendix A.

The model validation confirmed what was highlighted by the ROC curves. The best performance was offered by FT-MIR and, here too, the best classification was reached for Mis honeys, whatever the spectroscopic technique. Interestingly, in the case of FT-MIR, an overall improvement of the models’ prediction ability was achieved as more harvesting seasons were included in the training set, with all the binary models reaching CCRs >90% in the prediction of HN2017 (i.e., last step of the validation scheme). It must be pointed out that small differences (e.g., 0.1–0.2%) between the results have to be assessed with caution since these classification outcomes cannot be tested for statistical significance. Nevertheless, the overall trends have been clearly evidenced.

To further enhance the obtained results, the DF strategies described in Section 2.3.4. were attempted and the CCRs achieved in the HN2017 prediction summarized in Table 2.

All the DF methods provided satisfying classification performance, with HL-DF showing the highest scores, followed by ML-DF and LL-DF. The HL-DF reached comparable results respect to the FT-MIR (Table 1), with slightly better scores in the BA-Cat model and lower CCRs achieved in the BA-Mis honeys discrimination. A further attempt was made by combining the data blocks from two platforms only (i.e., FT-MIR+FT-Raman, FT-MIR+NIR and FT-Raman+NIR). However, no significant classification improvement was achieved (data not shown).

### 3.3. SIMCA Modelling

FT-MIR dataset underwent to SIMCA modelling as, in the light of the above results, it proved to be the most promising option for a hypothetical fingerprinting method for honey authentication. BA was set as target category to be modelled; thus, Cat and Mis samples represented the alien objects to be rejected by the model. Five PCs were considered sufficient for proper modelling as they accounted for >95% of the original data variance. The confidence level was set to α=0.05 and the classification rule was based on the so-called *T*^2^-*Q* augmented distances. The same year-by-year validation was adopted.

SIMCA results are reported as sensitivity, specificity and overall CCRs in Table 3.

Highly sensitive and specific models were produced, confirming what expected from the excellent classification previously obtained. In accordance with the PLS-DA results (Table 1), the inclusion of 2015 and 2016 harvest in the model training led to an overall enhancement of the model performance. Remarkably, within the prediction of HN2017, 39 out of 45 BA samples were correctly recognized as belonging to the target class (86.7% sensitivity), while 15 out of 17 Cat and 26 on 28 Mis honeys were rightly rejected by the model (91.1% specificity). *T*^2^ and *Q* distances of the predicted HN2017 samples are graphically represented in Figure 3.

## 4. Discussion

Since the analysis of all the collected honeys was carried out over a 4-years period, ensuring the absence of instrumental drift among the analysis batches was the first concern. The routine maintenance of the equipment, typically performed once a year, includes the substitution of overused components (e.g., source) and re-alignment of the interferometer, which may easily result in signal intensity (i.e., absorbance) shifts. Such technical variations, if not properly handled, may give rise to fingerprint deviations that prevent the use of the classification model for its ultimate purpose: the prediction of new harvests [24]. As an example, Woodcock et al. observed a clear separation between honey samples analysed in two consecutive years. However, the authors were not able to definitely attribute such trend to different sample’s characteristics, rather than the use of a non-standardized instrument [13].

Both the unsupervised and supervised chemometric approaches evidenced the presence of actual differences between the honeys having diverse provenance. Such differences are unlikely due to random variation or overfitting issues. In fact, it is worth stressing that the employed validation strategy allowed any developed model to be challenged with independent external test sets. With the main factor under investigation being the geographical origin, it is reasonable to ascribe the samples separation to the distinct environmental features of the three Argentinian regions. Variations of soil and weather conditions likely result in different melliferous floras foraged by the bees, which is known to have the greatest influence on the honey’s chemical composition [38]. Buenos Aires province is located within the ecoregion *Pampeana*, where the temperate climate and abundant rainfall encourage extensive crop cultivation. Misiones province is characterized by its typical flora, known as “Missionary Forest”, favored by the subtropical weather of the ecoregion *Selva Paranaense*. The peculiar characteristics of this ecoregion might underlie differences in the honeys’ physiochemical properties, which would explain the better results achieved in the classification of Mis samples. While Buenos Aires and Misiones regions show fairly uniform climate conditions, five different ecoregions are recognized in Catamarca (i.e., *Yungas*, *Chaco Seco*, *Monte de Sierras and Bolsones*, *Puna* and *Altos Andes*) and therefore a number of microclimates can be encountered, from the subtropical rains in the east, to the arid highland in the west [39]. Therefore, the larger overlap of Cat samples over the other classes might be due to this climate, and thus botanical, heterogeneity.

All the employed spectroscopic techniques provided more than satisfying performance, confirming the high potential of vibrational spectroscopy as rapid screening tool for honey authentication. Although lot of research has been done in the application of vibrational spectroscopy for honey testing, cross-platform comparisons have been scarcely documented. Tahir and co-workers observed equivalent performance of FT-MIR and FT-Raman spectroscopy for the prediction of phenolic compounds content and the antioxidant activity in honey [40]. Ballabio et al. recently evaluated five different technologies, including FT-MIR, NIR and FT-Raman spectroscopy, for the botanical origin identification of honeys [41]. The authors reported better classification provided by NIR, respect to FT-MIR and FT-Raman spectroscopy. Nevertheless, the same authors pointed out that such outcomes have to be assessed with caution due to the small size of the sample set. Within the present work, FT-MIR shown to be the best option for honey fingerprinting, providing always the largest AUC within the ROC curves, as well as superior CCRs (>90%) through the validation process. The reason probably lies in the better sensitivity and higher S/N normally provided by FT-MIR instruments respect to NIR and FT-Raman, since fundamental absorptions are being measured in the MIR region [42].

As pointed out in Section 3.2, LL-DF provided the poorest results among the attempted DF strategies. This is consistent with the literature, where LL-DF approach either did not produce substantial classification/prediction improvement over the single techniques or was outperformed by higher-level DF [40,41,43]. The explanation can be found in the high collinearity of vibrational spectroscopy data. In fact, LL-DF introduces, along with useful information, a large number of redundant and irrelevant variables. Such noise is, for example, reduced in the ML-DF by the features extraction prior to the concatenation. Concerning the ML-DF and HL-DF, despite the noteworthy results, no significant classification enhancement was reached respect to the FT-MIR only. On the basis of the present outcomes, the combination of vibrational spectroscopic data cannot be regarded as worthwhile as no evident advantage has been provided over the individual techniques. The authors attributed the ineffectiveness of DF to the lack of information orthogonality between the combined data sources, which is crucial for the successful application of DF [30].

When evaluated under conditions “closer” to a real authentication scenario, FT-MIR still yielded remarkable classification scores. The lower CCRs achieved by SIMCA respect to PLS-DA are not surprising as DA algorithms use information about the modelled classes to maximize the group differences, whereas OOC methods “do not know anything about existence of alternative classes or samples”. In fact, despite the widespread opinion that “PLS-DA may go further than SIMCA”, performance comparisons of these two algorithms are not even consistent as they employ diverse amounts of modelling information [21]. As mentioned in Section 1, DA algorithms are not suited for one-class problems where only one target category is modeled against a heterogeneous group of off-specification products [23]. For this reason, the authors believe that the SIMCA results (Section 3.3) are more representative of the potential performance of a routine screening method based on FT-MIR fingerprinting. The classification achieved in the HN2017 prediction can be considered excellent for a rapid screening platform and demonstrated that, under a proper characterization of the class of interest, FT-MIR spectroscopy can be a powerful tool for honey authenticity purposes.

In the authors’ opinion, the results herein obtained can be sensibly extended to problems of honey’s floral origin. In fact, botanical/varietal and geographical origin of food products are often treated as separate issues in food authenticity studies; however, they are highly correlated and hard to be considered individually, especially in the case of natural products such as honey. For example, distinct geographic areas do not only provide different climatic conditions affecting the accumulation of phytochemicals in pollen and nectar, but also normally offer diverse melliferous flora foraged by the bees. All these factors and relationships cannot be ignored in the development of methods for honey’s origin confirmation.

Despite the remarkable outcomes, in must be pointed out that the development of a comprehensive model able to identify the geographic origin of an unknown sample is unrealistic; it would require an exhaustive sampling of world honeys over several harvest years. Furthermore, honeys from different localities may not have unique spectral signatures due to similarities in vegetation. Thus, it is unlike to reach similar performance at a world-level. We believe that a fundamental knowledge of the limits and capabilities of the chosen methods is essential for their correct utilization and interpretation. Screening platforms based on spectroscopic fingerprints find the best applicability at a company-level, where the “boundaries” of the application can be clearly defined. Typical examples are internal quality assurance or the management of incoming raw materials from suppliers with established relationships. In these contexts, the target classes can be appropriately outlined and sampled in a representative way.

## 5. Conclusions

Honey authenticity remains a challenging issue to deal with as reliable and manageable methods for its floral and geographical origin confirmation are still lacking. Several feasibility studies have been reported in literature to demonstrate the capabilities of vibrational spectroscopy for the discrimination of honey’s botanical and/or geographical origin.

A key feature of the present work was the realistic and rather large variability included in the sample set. All the collected honeys were multifloral, thus covering differences in nectar sources. Besides, seasonal climate fluctuations were also considered by repeating the sampling over four consecutive harvesting seasons. This extra variation is of great benefit for the robustness of any developed model and crucial to demonstrate its capabilities under real-world conditions.

Excellent classification scores were achieved by all the technologies and the adopted validation strategy allowed to exclude any issue related to model overfitting. The nearly perfect classification results provided by FT-MIR suggested its major potential for honey fingerprinting. DF strategies yielded satisfying outcomes, however, no significant improvement in discrimination power was achieved respect to FT-MIR. Therefore, within an industrial context, a multi-platforms spectroscopic fingerprint is a choice that should be carefully evaluated from a cost/benefit standpoint. In fact, it must be considered that a multiple sample fingerprinting would represent an increased expense in terms of equipment and expertise, making the food control process more time and labour-demanding.

SIMCA modelling was successfully applied on the FT-MIR dataset and demonstrated that the use of large and representative training sets can definitely improve the model robustness over analytical and biological factors. The year-by-year validation not only ensured the results reliability, but also well reproduced a hypothetical quality control context where, reasonably, spectral libraries are gradually enlarged with newly recorded spectra. In the author’s opinion, such results can be considered a reliable performance estimation of a potential FT-MIR-based fingerprinting method.

## Figures and Tables

**Figure 1 foods-09-01450-f001:**
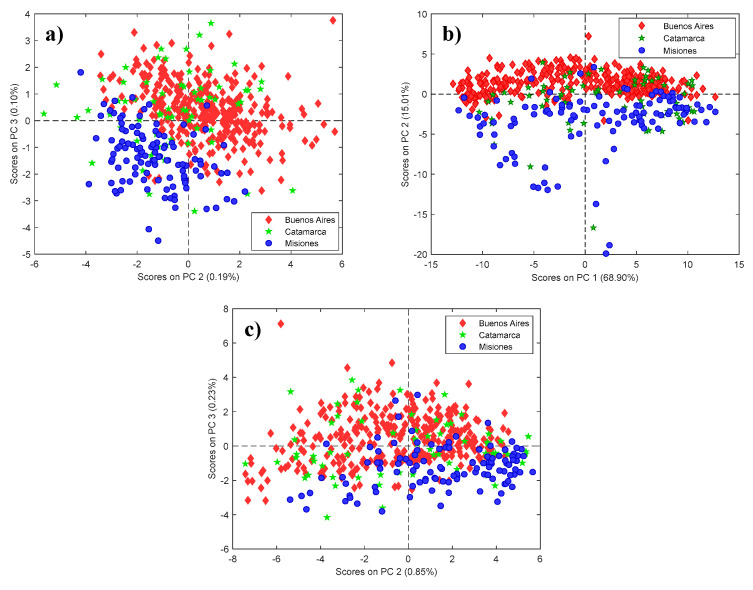
Scatter plot of PC1 vs PC2 scores obtained from FT-MIR (**a**), FT-Raman (**b**) and NIR (**c**) data. Objects are marked according to the provenance region (Red diamond: BA; Green star: Cat; Blue circle: Mis).

**Figure 2 foods-09-01450-f002:**
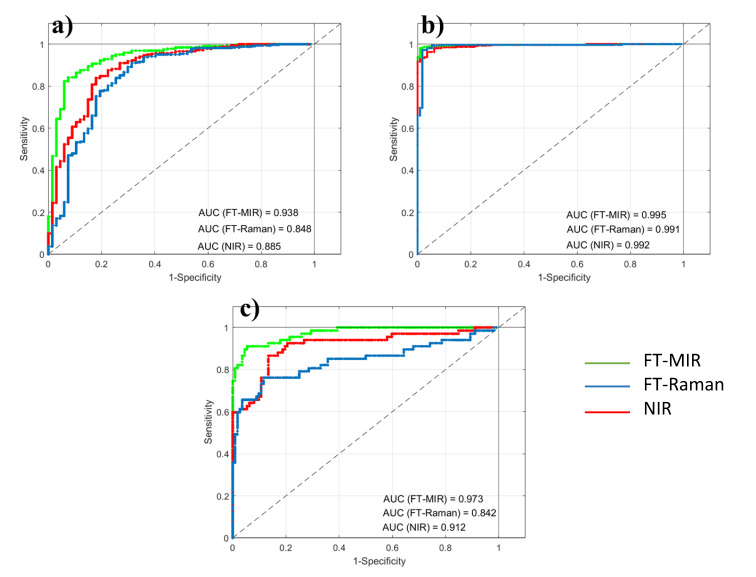
ROC curves related to the binary classification models (**a**) BA vs. Cat; (**b**) BA vs. Mis; (**c**) Cat vs. Mis.

**Figure 3 foods-09-01450-f003:**
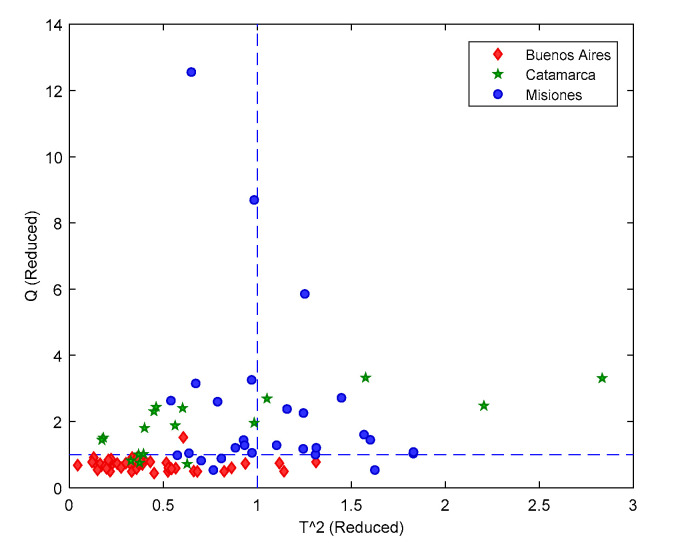
SIMCA modelling on FT-MIR data. Projection of HN2017 objects onto the *T*^2^ (reduced) vs. *Q* (reduced) model space of class BA.

**Table 1 foods-09-01450-t001:** PLS-DA prediction results expressed as correct classification rates (FT-MIR data).

Predicted Harvest	Correct Classification Rate (%)		
	BA vs. Cat	BA vs. Mis	Cat vs. Mis
2015	84.6	88.4	92.3
2016	91.8	100.0	92.5
2017	91.9	100.0	95.5

**Table 2 foods-09-01450-t002:** PLS-DA classification results of HN2017, expressed as correct classification rates, according to the different DF strategies.

Predicted Harvest	Correct Classification Rate (%)		
	BA vs. Cat	BA vs. Mis	Cat vs. Mis
LL-DF	85.4	91.7	80.0
ML-DF	87.0	98.6	80.0
HL-DF	93.5	98.6	95.5

**Table 3 foods-09-01450-t003:** SIMCA modelling results of class BA (FT-MIR data) according to the different harvesting seasons, expressed as sensitivity, specificity and overall correct classification rates.

Predicted Harvest	Sensitivity (%)	Specificity (%)	CCR (%)
2015	61.0	89.7	69.6
2016	90.6	75.0	85.8
2017	86.7	91.1	88.8

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
