# Peer review of "Vibrational Spectroscopy Coupled to a Multivariate Analysis Tiered Approach for Argentinean Honey Provenance Confirmation"

_foods, 2020, doi:10.3390/foods9101450_

Round 1

Reviewer 1 Report

The authors demonstrate the FT-IR,  Raman, and Near-IR spectral data have sufficient specificity to differentiate between three authenticated honey sample sets.  Although chemomarkers most likely also are present because the organoleptic properies of the honey sample sets are a priori known to be different,  the spectroscopic evidence is that the spectral fingerprint is different as well.   Thus spectral lines unique between the sample sets are essentially chemometric identification of what appears to be chemically unique site to site.

Physically separating some chemical from a wide diversity of chemical (e.g. by chromatographic) techniques is not required.

The data in Table 1 contains three paired analytical results :  91.8 and 91.9; 100.0 and 100.0;  92.3 and 92.5.  Do the authors claim that a difference of 0.1 or 0.2% is real?  No error bars are included with the numbers, so readers of the article cannot tell if the numbers are correct to  1%, 0.3% or  0.1% for example. The numbers 100.0 and 100.0%  reported assumes an accuracy of +/- 0.1%

Accuracy and precision need to be reported  99. , or 99.9 ,  or 99.99%.  

This said, the authors need to explain why they see so little uncertainty in this set of analytical results.

BA-Mis results from Figure 1 clearly shows Buenos Aires and Misiones honey (red and blue) are readily distinguishable from each other. Catamarco (green) results are more intermixed / intermingled.

Either Table 1 needs to be deleted,  replaced by a different table, or the results explained in a way that contain measurable variance among the data sets.

Author Response

Please, find attached a word file for a full rebuttal of the comments moved by Reviewer#1

Reviewer 2 Report

The article describe the spectroscopic analysis of Argentinian honeys for provenance discrimination. The paper is very well written and all results are clearly described in text.

Comments:

Lines 139-143: Information about laser energy is missing              

Chapter 3.1: Infrared spectra looks very consist without any extreme drift of baseline. Also, after data pre-processing the Score plots looks very nice and segregation of Buenos Aires and Misiones groups is evident. On the contrary, the data from Raman analysis show high drift of baseline that could be affected by fluorescence (even if at laser 1064 nm it is partially reduced). Did authors try baseline correction? The segregation in PCA score plot could be better and clearer.  

Fig. 1: Did authors try 3D score plot? The third component could show the segregation of Catamarca honey samples from others.

PLS-DA was applied on dataset and no significant classification was found. Why are not data shown? Those should be at least in Supplement. Authors applied PLS-DA for comparison only of two groups of honey samples. If authors will compare whole dataset (i.e. three groups) and then will apply variable importance in projection, they can get the most significant regions for segregation. After “noise” elimination it could be re-visualised in PCA.

The main information missing in manuscript is description of main signals/region (even if it should be only fingerprint method) that are responsible for segregation of honey samples. In line 287-288 is some information about regions, but no information if it is from statistical evaluation (PCA do not give you this kind of information) or it is just set by authors. Without this information is manuscript unusable in practise because anyone cannot apply this method and must start their research from beginning.

Author Response

Please, find attached a word file for a full rebuttal of the comments moved by Reviewer#2

Round 2

Reviewer 2 Report

The manuscript was improved and I recommend it for publication.